# The Influence of Antioxidant Plant Extracts on the Oxidation of O/W Emulsions

Arielle Springer [1,*], Helena Ziegler [1,2] and Katrin Bach [2]

1   Fraunhofer Institute for Process Engineering and Packaging IVV, Giggenhauser Str. 35, 85354 Freising, Germany
2   MCI Management Center Innsbruck, Universitaetsstrasse 15, 6020 Innsbruck, Austria
*   Correspondence: arielle.springer@ivv.fraunhofer.de

**Abstract:** The demand for natural cosmetics has steadily increased in recent years. However, challenges occur especially in quality preservation regarding oxidative spoilage of natural cosmetic products, as the use of synthetic preservatives and antioxidants is limited. Therefore, it is important to find nature-based ingredients to ensure shelf life in natural cosmetic formulations. As a result, potential is seen in the use of plant-based antioxidant extracts. The aim of this work was to determine the suitability of the method combination by measuring the antioxidant activity, oxygen concentration, and volatile oxidation products via gas chromatography (hexanal) for the characterization of the influence of some plant extracts on the oxidative stability of natural cosmetic emulsions. Plant extracts of Riesling (*Vitis vinifera*) pomace, apple (*Malus domestica*) pomace, coffee (*Coffea arabica*) grounds, cocoa (*Theobroma cacao*) husk, and coffee (*Coffea arabica*) powder extract were incorporated in stable O/W emulsion formulations, while an emulsion without extract functioned as blank. Afterwards, the emulsions were subjected to 3-month accelerated storage tests with and without light exposure. Their oxygen uptake was investigated, and headspace gas chromatography measurements were performed to detect the fatty acid oxidation products formed during oxidative processes in the samples. The results showed that all emulsion samples under light exposure had a higher oxygen uptake and an increase in the characteristic fatty acid oxidation products compared with those stored under light exclusion. However, differences in oxygen uptake under light exposure were observed depending on the plant extract. Therefore, for O/W emulsions, the daily oxygen consumption rate correlated exponentially with the antioxidant activity, and the hexanal concentration correlated linearly with the daily oxygen consumption rate.

**Keywords:** oxidation kinetics; oxygen concentration; oxygen uptake; gas chromatography; mass spectrometry; volatiles; VOCs; pro-oxidant; secondary plant substances; SPS; shelf life

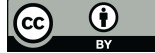

## 1. Introduction

In recent years, an increased consumer awareness for an environmentally friendly purchasing behavior, sustainable packaging, and a preferred "green" purchasing behavior is evident [1]. Therefore, the cosmetics industry has strongly focused on consumer demand for certified natural cosmetics in combination with a healthier lifestyle [2,3]. At the same time, in line with the increased demand for certified natural cosmetics, sales for natural cosmetic products in Germany had continuously increased from 2007 to 2019 [4]. For example, natural certified skin care products often contain vegetable oils, due to their effectiveness in the dermal treatment or relief of dry skin, atopic eczema, and psoriasis. In particular, the essential unsaturated fatty acid $\gamma$-linolenic acid is metabolized in skin lipids to form intracellular ceramides and prostaglandins, which increase skin hydration and have anti-inflammatory effects [5–10].

However, natural oils are more susceptible to oxidative processes than synthetic emollients due to the content of unsaturated oils and the possible impurities from plant

ingredients [11]. Especially, natural cosmetic products containing them are prone to oxidation, as additionally, the use of synthetic antioxidants in natural cosmetics produced under the labels of ECOCERT, NATRUE, or COSMOS is prohibited. These result in undesirable rancid aroma notes, quality losses of the products, a limited shelf life, and decreasing consumer acceptance [12–15]. In most cases, fat oxidation occurs between polyunsaturated lipids and oxygen in the form of decomposition and polymerization reactions and leads to degradation products, such as peroxides, alcohols, aldehydes, lactones, alkanes, methyl ketones, and carboxylic acids [16]. As the oxidation rate as well as the rapidity of the induction period increases, the more double bonds are present in a fatty acid molecule. Overall, the process of autoxidation is influenced by various factors, such as the temperature, the composition of fatty acids, the concentration and effectiveness of pro- and antioxidants, the oxygen partial pressure, and oxidation catalysts, such as heavy metal ions such as copper and iron from pigments, natural raw materials, or the equipment in the production process. Photo-oxidation is induced by photosensitizers in the presence of light or higher energy electromagnetic radiation and leads to an acceleration of autoxidation [11,17,18].

Even more than pure oils, emulsions are susceptible to oxidation processes. Depending on the emulsion type, differences for the occurrence of lipid oxidation are caused by the different oil and water phase distribution. Basically, the oxidation behavior in emulsions is an interplay of many factors, such as the presence of pro-oxidants, the droplet characteristics, oxygen concentration, interactions with aqueous phase components, and interfacial characteristics (e.g., electrical charge, physical and chemical barrier). Lipid oxidation in water-in-oil (W/O) emulsions is associated with a similar lipid oxidation behavior like bulk oils due to the larger oil phase that is exposed to direct air contact [19]. In contrast to this, in oil-in-water (O/W) emulsions, lipid oxidation occurs due to the large surface of the dispersed phase that allows dissolved oxygen from the aqueous phase to reach the dispersed oil phase. In general, the oxidation mechanism in O/W emulsions is based on the lipid hydroperoxide reactions with pro-oxidants from the aqueous phase, which leads to radical formation at the surface of the micelles. Consequently, this initiates the oxidation of fatty acids inside the micelle and finally leads to the production of undesirable rancid compounds. It is necessary that these unsaturated fatty acids are protected so that they can maintain their valuable effects on the skin and their oxidation products do not negatively affect the bulk [19–21].

To counteract this, potential is seen in the addition of natural plant antioxidants for quality preservation in natural cosmetics [22,23]. Secondary plant substances contained therein are known for their ability to limit and thus interrupt the radical chain reactions of oxidative degradation by transferring hydrogen atoms or electrons or by forming complexes with metal ions or reduction of oxygen concentration. It has already been shown that secondary plant substances can have up to 16-fold higher antioxidant potential than synthetic antioxidants [24]. However, these properties depend largely on the source of the extracts and the processing procedure. Especially, by-products of food production show various challenges that should be taken into account to preserve the quality of active ingredients, such as polyphenols and unsaturated fatty acids. Nevertheless, their use in cosmetics seems to be beneficial [25].

To assist in choosing the right extract, there are numerous studies characterizing the antioxidant effect of plant extracts in emulsions using 2,2-diphenyl-1-picrylhydrazyl-(DPPH) assay [26–28]. However, this antioxidant effect often refers to the effect on the skin or physical stability, not to the oxidative shelf life extension in the product. Some studies investigated the effect on the quality changes, but they considered only specific plant raw materials, not their correlation with the assay results [29–32]. Other studies analyzed the oxidation of O/W emulsions depending on the plant extract using peroxide value. As a result, these studies showed a positive impact of the plant extracts on the oxidation stability in emulsions and the potential of these extracts as an alternative for synthetic antioxidants against lipid oxidation [29,33–35]. However, lipid hydroperoxides degrade with time, leading to volatile off-flavors but a lower peroxide value, resulting in a potential missed

correlation between oxidation and peroxide value results with longer storage periods [11]. Another parameter to determine the oxidation in emulsion are thiobarbituric acid reactive substances (TBARS), as already used in the literature for the quantification of oxidation in O/W emulsions [36–38]. Nevertheless, TBARS as a sum parameter might not be suitable for detecting specific fat oxidation products, since thiobarbituric acid only occurs in specific fat degradation reactions and fatty acid spectra. The color reaction used to detect TBARS might be highly dependent on the matrix and could be distorted by cosmetic coloring agents. Although it correlates with oxidation, it does not provide any information about the individual substances and their influence on quality [39]. Another commonly used method is rancidification, where the oxidation of emulsions was measured via Rancimat in several publications [40,41]. However, one disadvantage of this method might be the absence of information about the individual aroma substances and their sensory influences. Additionally, the information gained from Rancimat measurements cannot be transferred to calculate the shelf life or packaging barrier requirements of the products since the amount of absorbed oxygen is not quantified. Furthermore, there is also literature describing fat oxidation in food emulsions using all three methods: measurement of peroxide value, TBARS, and Rancimat [42]. Although these measurements were appropriate for the estimation and comparison of oxidation processes, they also delivered no further information about shelf life and packaging. Measurements of the fatty acid oxidation products (ketones, aldehydes) via gas chromatography (GC) might realistically represent the shelf life process by the identification and quantification of volatile fat oxidation products, even below the human sensory detection limit. Hexanal can be used as a representative volatile substance for fat oxidation, as it is formed from omega-6 fatty acids such as linoleic acid, which is present in rapeseed oil at amounts of 15–30% [11,43]. The advantage of a headspace (HS) analysis is the separation of the matrix and the consideration of only those substances that volatilize from the matrix on their own. Mass spectrometry (MS) can also provide a more precise identification of the substances using databases.

So far, no studies were found that focused specifically on the investigation of fatty acid oxidation in cosmetic O/W emulsions depending on the addition of plant extracts and the type of emulsion by determining the oxygen uptake, the daily oxygen consumption rate, and the fatty acid oxidation products formed.

Thus, the aim of this study was to determine the suitability of the method combination by measuring the antioxidant activity, oxygen concentration, and volatile oxidation products via gas chromatography for the characterization of the influence of some plant extracts with antioxidant activity on the oxidative stability of natural cosmetic O/W emulsions. This should be investigated by combining different analysis methods, such as measuring of the antioxidant activity, the oxidation kinetics, and quantification of hexanal as a fatty acid oxidation product via HS-GC–MS measurements.

## 2. Materials and Methods

### 2.1. Preparation of the Plant Extracts

To prepare the plant extracts, the following raw materials were used: pomace of Riesling grape (*Vitis vinifera*), mixed apple pomace (*Malus domestica*), brewed grounds of coffee (*Coffea arabica*), cocoa husk (*Theobroma cacao*), fresh coffee powder (*Coffea arabica*). The raw materials were freeze-dried and ground to 0.5 mm particle size. It is known that the extraction process has a strong influence on the antioxidant effect of the extract, which is why the same method was used for all samples in this study [44]. The plant extracts were prepared using the following extraction method: 70% ethanol, solid-to-liquid ratio (S/L) 1/10, 60 min at room temperature. The dry matter of extracts was determined using an MA100 infrared moisture analyzer (Sartorius AG, Göttingen, Germany).

### 2.2. Quantification of the Antioxidant Activity (In Vitro)

The antioxidant activity of the extracts was determined using the oxygen radical absorbance capacity (ORAC) assay according to Huang et al. [45] and Platzer et al. [46].

A stock solution was prepared by dissolving 1.33 mg fluorescein in 75 mM phosphate buffer, which was then stored in the dark at 4 °C. Based on this, the working solution was prepared by a 1:1000 dilution with 75 mM phosphate buffer. In addition, 414 mg of 2,2′-azobis(2-amidinopropane) dihydrochloride (AAPH) was dissolved in 10 mL of 75 mM phosphate buffer and also stored at 4 °C until use. Subsequently, 25 µL of each sample containing 150 µL of fluorescein working solution was added to the wells of a 96-well plate, and the decrease in fluorescence was measured for 30 min at 37 °C after the reaction was started by adding 25 µL of AAPH solution. Fluorescence was recorded every 60 s as kinetics versus time in a microplate reader (Infinite 230 PRO, TECAN (Männedorf, Switzerland)) at an excitation wavelength of 485 nm (20 nm bandpass) and an emission wavelength of 535 nm (20 nm bandpass) (s). The raw data were analyzed according to Platzer et al. [46].

### 2.3. Preparation of the O/W Emulsions

To prepare the O/W emulsions in a single production, the following ingredients were used: AQUA, BRASSICA CAMPESTRIS SEED OIL, ETHANOL, GLYCERYL STEARATE, EXTRACT, CITRIC ACID, XANTHAN GUM. All formulations were prepared in 150 g samples. The oil phase was set to 22.8%, the emulsifier to 7.0%, the thickener to 0.2%, and the pH regulator to 0.2% for comparability between the samples. The basic formulation can be found in Table 1.

**Table 1.** Overview of the formulations used.

| Ingredient | Step | Content (g/100 g) |
|---|---|---|
| Canola (*Brassica napus)* Oil | 1 | 22.8 |
| Xanthan Gum | 1 | 0.2 |
| Glyceryl Stearate SE | 2 | 7.0 |
| Plant Extract | 2 | See Table 2a |
| Added Ethanol (96%) | 2 | See Table 2b |
| Citric acid | 2 | 0.2 |
| Water | 2 | Add to 100 |

**Table 2.** Calculation of the plant extract content.

| Ingredient | Dry Mass in Extract (%) | a: Extract Content in Emulsion (g/100 g) | Ethanol in Plant Extract (g/100 g) | b: Content Added Ethanol (g/100 g) |
|---|---|---|---|---|
| Riesling (*Vitis vinifera*) pomace extract | 3.41 | 5.27 | 3.69 | 11.78 |
| Apple (*Malus domestica*) pomace extract | 4.75 | 3.79 | 2.65 | 12.86 |
| Coffee (*Coffea arabica*) grounds extract | 0.78 | 20.51 | 14.36 | 0.67 |
| Cocoa (*Theobroma cacao*) husk extract | 0.89 | 20.23 | 14.16 | 0.88 |
| Coffee (*Coffea arabica*) powder extract | 1.58 | 11.39 | 7.98 | 7.32 |

Plant extracts are natural products and may vary in their dry matter content. Assuming that the oxidatively affecting active ingredients were located in the dry matter, the input of the extracts had to be normalized accordingly to achieve a comparability between different extracts and correlatability with the analysis data. The concentration of the extracts in the emulsion was set to $0.16\% < c_{(DM)} < 0.18\%$ of their dry mass. Consequently, the input concentrations of the extractive solution varied. The ethanol content was set to 15%. Microbial preservation was necessary to clearly attribute the consumption of oxygen to oxidation rather than microbial metabolism. Ethanol has a very good preservative effect with a broad pH action spectrum, which is due to its nonspecific protein denaturation effect against all types of microorganisms [47]. Additionally, ethanol showed the smallest effect itself on oxidative processes in previous studies [48]. As the extract was provided as

a 70% ethanolic solution, the water fraction and the ethanol fraction had to be calculated separately to reach 15% ethanol in the aqueous phase. The additional ethanol was added as a 96% solution. The calculation of the plant extract and ethanol concentrations is shown in Table 2.

In the first step (1), the xanthan gum was well suspended in the oil phase (weight in Table 1). In the next step (2), glyceryl stearate (SE), ethanol, plant extract, citric acid, and water were added, and the mixture was tempered (weight in Tables 1 and 2). Then, the homogenization process was carried out with the high-performance disperser, first at 10,500 rpm for 60 s, and then at 17,800 rpm, also for 60 s. It is important to work in circular motions during this process and at the same time to distribute the ingredients evenly with a spatula. After the homogenization process, the mixture was cooled to room temperature. The finished mixture was homogenized again with the aid of the disperser at 10,500 rpm for 60 s and afterwards at 17,800 rpm, also for 60 s. The product was vacuumed at 300 mbar for 1 min to remove air bubbles that could distort oxygen uptake from the headspace.

### 2.4. Quantification of the Oxygen Uptake and Oxidation Products

In order to determine the oxidation kinetics and products, oxygen concentration was measured during storage for 3 months under light exposure (halogen retail lighting, approx. 1000 lux, 6000 K, OSRAM Lumilux Cool Daylight, 30 W/865) as well as light exclusion (dark climatic cabinet) for all variants. The experimental measurement of the oxygen partial pressure of the headspace gas was carried out via chemical optical sensors and further developed according to the literature [48]. The sensors were covered with a small aluminum tape to exclude the sensor surface from the light exposure. The chosen storage conditions might represent the first 5–6 months after production. Oxygen uptake can also be represented as a function of time as daily oxygen consumption rate in mg oxygen per 100 g per day. As a simple numerical value, the daily oxygen consumption rate is easier to correlate with other data. In addition, plotting the daily oxygen uptake rate as a bar graph allows better differentiation of small differences between samples [49]. At the end of the storage, a gas analysis was performed. Microbial metabolites, such as $CO_2$, were below the detection limit of 0.1%.

The quantitative determination of hexanal was carried out using the HS GC–MS device Shimadzu GCMS-QP2010 Ultra. A customized method development was performed to better separate the numerous peaks from the extracts. Table 3 lists the measurement parameters, whereas the experimental setup is shown in Figure 1. For quantification of hexanal in the stored samples, we calibrated with 0, 1, 2, and 4 mg/L hexanal diluted from 100 mg/L hexanal in DMSO (0, 40, 80, and 160 μL added to 4 g emulsion). The calibration was performed in matrix (sample without extract) and gave an accuracy of $R^2 = 0.9991$.

### 2.5. Data Analysis

Statistical analysis of the measurement results was performed by the one-way variance method (ANOVA) with all significant decimals using SigmaPlot (Systat Software, San Jose, CA, USA), which corresponds to an unpaired t-test. If there was a significant difference, an additional paired test was performed using the Holm–Šidák method [50,51]. This method is used to counteract the problem of multiple comparisons while providing familywise error control that is exact for tests that are stochastically independent. The significance level for all tests was $p > 0.05$.

To determine the relationship between the antioxidant activity of the extracts (ORAC assay results), the daily oxygen consumption rate, and hexanal concentration, a correlation analysis was carried out by plotting on an xy-chart and calculating an equalization function based on a causal connection of the reaction mechanisms of the occurring processes. The higher the coefficient of determination $R^2$, the higher the correlation between the data. An exponential function was used for the correlation between ORAC assay results and daily oxygen consumption rate due to the evaluation method of the ORAC assay as a measurement of the radical absorbance capacity. The ORAC value takes into account the

complete course of the chemical reaction, which makes it possible to include antioxidants that prolong the lag time and antioxidants that slow down the reaction rate [52,53]. For the correlation of the daily oxygen consumption rate and the hexanal concentration, a linear relationship is used, because one molecule of hexanal is caused by the degradation of the fatty acid reacting with one molecule of oxygen [17]. In both correlations, the data of the samples without extract were excluded, since they contain a different amount of oxidizable substances than those with extract. To determine a correlation, an increase in daily oxygen consumption rate and hexanal concentration should occur. Since the samples stored in darkness did not show significant oxidation, a correlation analysis was not carried out.

**Table 3.** Parameters of the HS GC–MS method.

| Parameter | Value |
|---|---|
| Headspace conditioning | 30 min 60 °C |
| GC column | Optima WAXplus, Thickness of 0.25 μm, diameter of 0.25 mm, length of 30 m, polar |
| Column oven temperature | 35 °C |
| Carrier gas | Helium 1.4 mL/min |
| Temperature gradient | |
| Ionization | Electron impact ionization, ion source temperature of 200 °C, 70 eV |
| Mass spectrometry | Detection: quadrupole, interface temperature of 230 °C, scan mode of 35–350 m/z, speed of 1666/min |

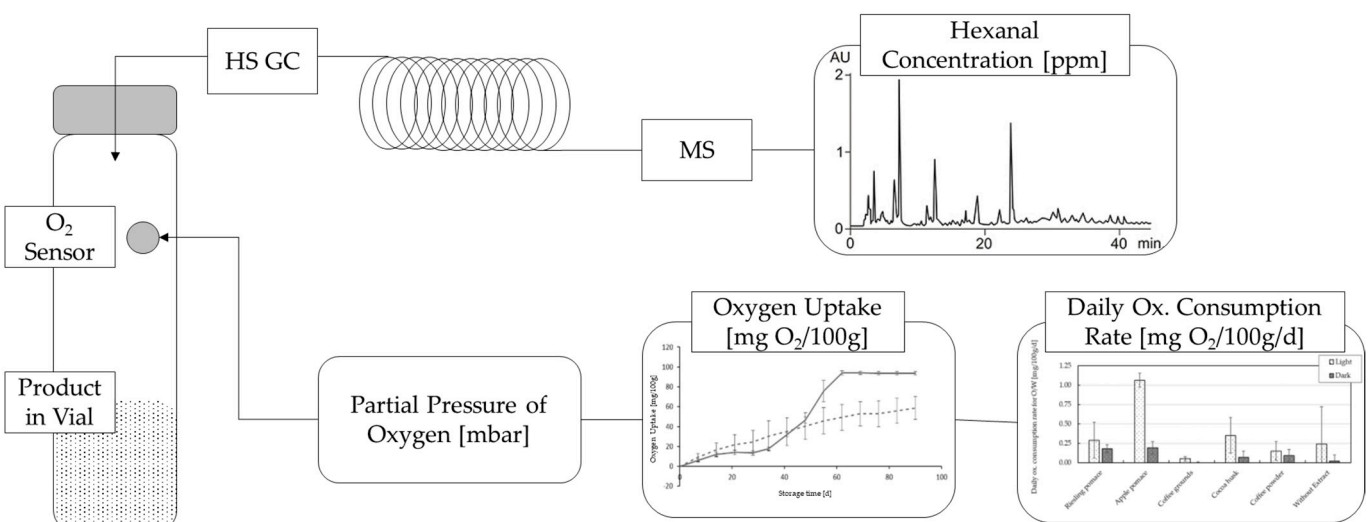

**Figure 1.** Scheme for the experimental setup for measuring the oxidation kinetics and oxidation products of emulsions during an accelerated storage test.

## 3. Results

### 3.1. Determination of In Vitro Antioxidant Behavior of Plant Extracts

In vitro antioxidant activities of the plant extracts were measured using the spectrophotometric ORAC assay. The results are shown in Figure 2. The values of the area under the curve (AUC) to dry mass (% DM) ratio varied from $(2.3 \pm 0.2) \times 10^7$ to $(2.3 \pm 0.1) \times 10^8$, representing a variation of one order of magnitude. The highest radical scavenging activity was reached by coffee ground extract $((2.3 \pm 0.1) \times 10^8)$, followed by coffee powder $((1.5 \pm 0.1) \times 10^8)$, cocoa husk $((8.7 \pm 0.3) \times 10^7)$, and Riesling pomace $((6.7 \pm 0.1) \times 10^7)$. Apple pomace reached the lowest result of $(2.3 \pm 0.2) \times 10^7$ in our test.

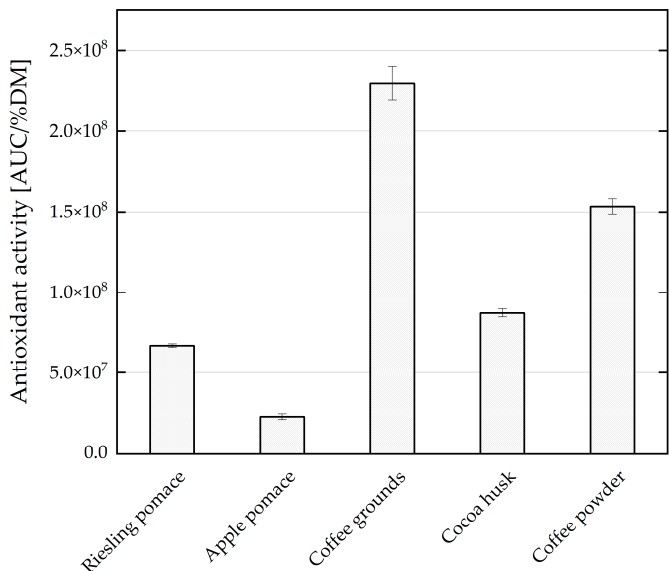

**Figure 2.** Antioxidant activity of extracts measured with ORAC assay.

The differences in the mean values among the treatment groups were greater than would be expected by chance. There was a statistically significant difference ($p < 0.01$). All samples were systematically tested among each other and differed significantly, as shown in Table 4.

**Table 4.** $p$-Values of all pairwise multiple comparison procedures (Holm–Šidák method) of the antioxidant activity of extracts measured with ORAC assay (significance *** $p < 0.01$).

|  | Riesling Pomace | Apple Pomace | Coffee Grounds | Cocoa Husk |
|---|---|---|---|---|
| Apple pomace | <0.001 *** |  |  |  |
| Coffee grounds | <0.001 *** | <0.001 *** |  |  |
| Cocoa husk | <0.001 *** | <0.001 *** | <0.001 *** |  |
| Coffee powder | <0.001 *** | <0.001 *** | <0.001 *** | <0.001 *** |

### 3.2. Oxygen Uptake of O/W Emulsions with Plant Extracts

Oxygen uptake was calculated from the oxygen concentration measured in the headspace of the samples. The following figures show the progression of oxygen uptake in mg $O_2$ per 100 g for all O/W emulsions with the added plant extracts stored at 30 °C in a heating cabinet under light exposure (Figure 3) as well as light exclusion (Figure 4) over a period of 90 days.

The oxygen uptake was in general higher for the samples stored under light exposure than in darkness over the entire storage time of 90 days. The higher oxygen uptake under light exposure (Figure 3) indicates increased oxidative processes in the presence of light. In the dark, all samples behaved similarly and showed low oxidation with a slight increase.

Since the oxygen uptake of the samples stored in darkness (Figure 4) did not significantly increase and did not show significant differences between the samples, it can be assumed that barely any oxidation occurred here. In particular, oxygen uptake increased in the reference sample without extracts after 60 days of storage in light and was then in the low range. The samples with Riesling pomace, coffee grounds, cocoa husk, and coffee powder behaved similarly. Oxygen uptake rapidly increased in the sample with apple pomace after the first 10 days of the storage and then remained after 60 days on a plateau that was about three times higher than other samples. This can be explained by an intensive radical chain reaction up to the complete absorption of oxygen.

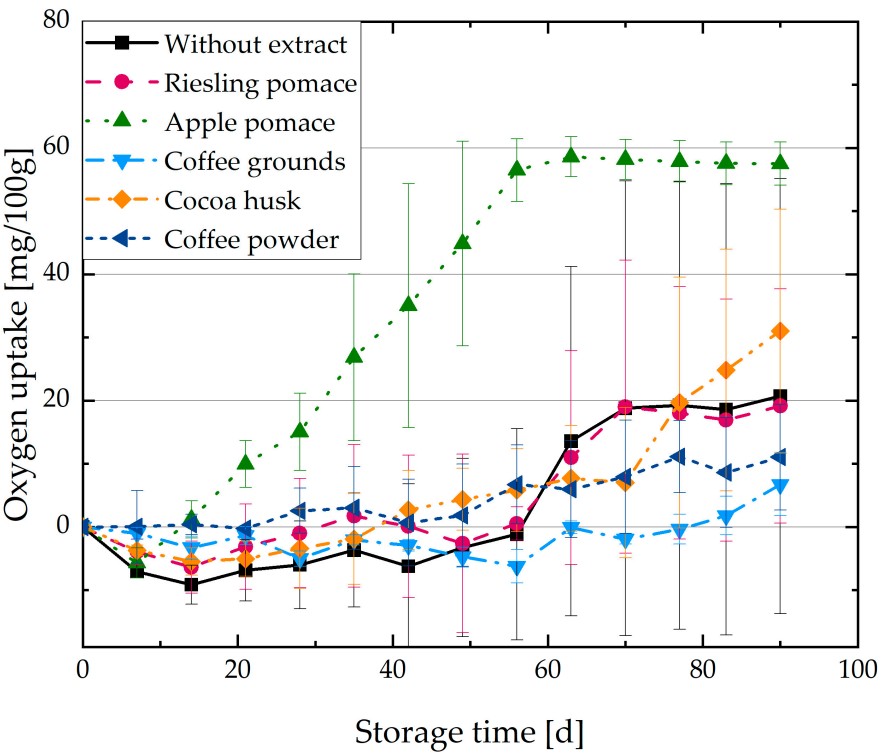

**Figure 3.** Oxidation kinetics of natural cosmetic emulsions with plant extracts stored in light. The figures show the oxygen uptake in mg $O_2$ per 100 g as a function of the storage time at 30 °C for the O/W emulsions with and without extract, as mean values (*n* = 3) with standard deviation (SD).

*3.3. Daily Oxygen Consumption Rate of O/W Emulsions with Plant Extracts*

The daily oxygen consumption rate was calculated from the oxygen uptake. It is shown as a bar chart in Figure 5 for the emulsions with Riesling pomace, apple pomace, coffee grounds, cocoa husk, and coffee powder extract as well as without extract stored in light exposure and exclusions.

The daily oxygen consumption rate of the emulsions with extracts stored in light was plotted as a function of antioxidant activity of the pure extracts measured with the ORAC assay and compared with an equalization function (Figure 6). The correlation of the O/W emulsions with extracts stored in light with the antioxidant activity of the extracts showed a high accordance of $R^2$ = 0.84 to an exponential graph. With increasing ORAC assay result, the daily oxygen consumption rate decreased.

The differences in the mean values of daily oxygen consumption rate among the treatment groups were greater than would be expected by chance. All samples were systematically tested among each other using the Holm–Šidák method (Table 5). It is clear that the emulsion with apple pomace had significantly higher values for daily oxygen consumption rate than the other samples when stored in light. The other samples did not differ significantly from each other.

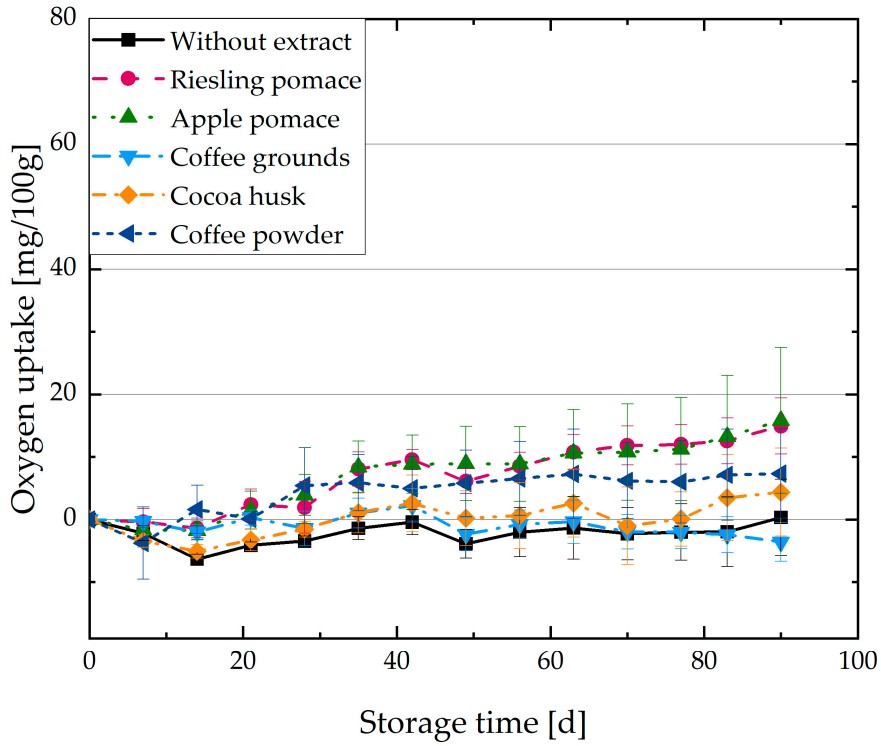

**Figure 4.** Oxidation kinetics of natural cosmetic emulsions with plant extracts stored in darkness. The figures show the oxygen uptake in mg $O_2$ per 100 g as a function of the storage time at 30 °C for the O/W emulsions with and without extract, as mean values (*n* = 3) with standard deviation (SD).

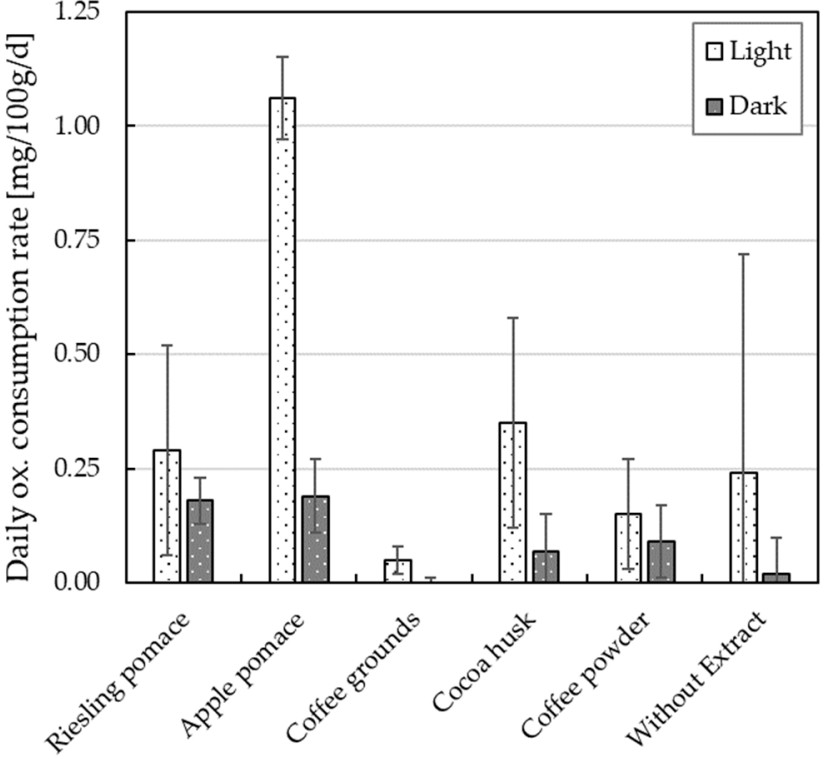

**Figure 5.** Daily oxygen consumption rate of O/W emulsions with plant extracts. The figure shows the daily oxygen consumption rate in mg $O_2$ per 100 g per day as a function of a storage time of 90 days at 30 °C for the O/W emulsions with and without extract in light and darkness, respectively, as mean values (*n* = 3) with standard deviation (SD).

### 3.4. Hexanal Concentration of O/W Emulsions with Plant Extract

Hexanal was measured with HS GC–MS after storage. The concentrations of hexanal in mg/L after 90 days are shown in Figure 7 for the emulsions with Riesling pomace, apple pomace, coffee grounds, cocoa husk, and coffee powder extract as well as without extract stored in light exposure and exclusions. In this graph, it is noteworthy that the concentration of the fat oxidation product hexanal is comparably high in the sample with apple pomace and the one without extract stored in light, even though the results previously showed a lower oxygen uptake and daily oxygen consumption rate. All other extracts and the samples stored in darkness were clearly lower.

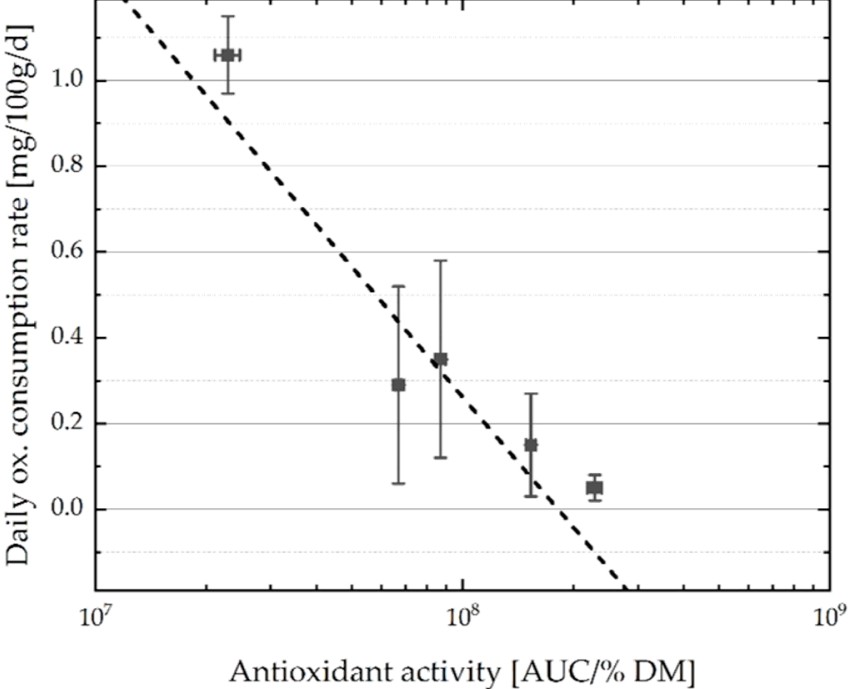

**Figure 6.** Correlation between daily oxygen consumption rate and antioxidant activity of emulsions with plant extracts. The figure shows the daily oxygen consumption rate in mg $O_2$ per 100 g per day as a function of the antioxidant activity in AUC/% dry mass measured with ORAC assay, as mean values ($n = 3$) with exponential equalization function ($R^2 = 0.84$) for all samples stored in light.

**Table 5.** *p*-Values of all pairwise multiple comparison procedures (Holm–Šidák method) of the daily oxygen consumption rate in O/W emulsions with plant extracts (significance *** $p < 0.01$).

| | | Riesling Pomace | | Apple Pomace | | Coffee Grounds | | Cocoa Husk | | Coffee Powder | | Without Extr. |
|---|---|---|---|---|---|---|---|---|---|---|---|---|
| | | **Light** | **Dark** | **Light** | **Dark** | **Light** | **Dark** | **Light** | **Dark** | **Light** | **Dark** | **Light** |
| Riesling pomace | Light | | | | | | | | | | | |
| | Dark | 1.000 | | | | | | | | | | |
| Apple pomace | Light | 0.001 *** | <0.001 *** | | | | | | | | | |
| | Dark | 1.000 | 0.946 | <0.001 *** | | | | | | | | |
| Coffee grounds | Light | 0.997 | 1.000 | <0.001 *** | 1.000 | | | | | | | |
| | Dark | 0.915 | 1.000 | <0.001 *** | 1.000 | 1.000 | | | | | | |
| Cocoa husk | Light | 1.000 | 1.000 | 0.004 *** | 1.000 | 0.939 | 0.649 | | | | | |
| | Dark | 0.999 | 1.000 | <0.001 *** | 1.000 | 0.999 | 1.000 | 0.974 | | | | |
| Coffee powder | Light | 1.000 | 1.000 | <0.001 *** | 1.000 | 1.000 | 1.000 | 1.000 | 1.000 | | | |
| | Dark | 1.000 | 1.000 | <0.001 *** | 1.000 | 1.000 | 1.000 | 0.989 | 1.000 | 1.000 | | |
| Without Extract | Light | 1.000 | 1.000 | <0.001 *** | 1.000 | 1.000 | 0.990 | 1.000 | 1.000 | 1.000 | 1.000 | |
| | Dark | 0.983 | 1.000 | <0.001 *** | 1.000 | 0.999 | 1.000 | 0.847 | 1.000 | 1.000 | 1.000 | 0.999 |

The results of the hexanal concentrations of all samples were tested systematically among each other using the Holm–Šidák method. The differences in the mean values among the treatment groups were greater than would be expected by chance (shown in Table 6). In this case, the sample with apple pomace extract also showed significantly higher values, but in addition, the sample without extract had also a significant higher hexanal concentration. The other samples did not differ significantly from each other.

The correlation between hexanal concentration as a function of the daily oxygen consumption rate for storage under light exposure and all extracts is shown in Figure 8. For the O/W emulsions, a correlation ($R^2 = 0.99$) was found between daily oxygen consumption rate and hexanal concentration, which indicates that the data are related. The higher the daily oxygen uptake rate, the higher the hexanal concentration after 90 days of storage.

Additionally, the correlation between hexanal concentration after storage and ORAC assay results as the antioxidant activity of the extracts was also tested (Figure 9). The O/W emulsion showed a relationship between the data under light exposure ($R^2 = 0.96$), where a lower antioxidant activity of the extract led to a higher hexanal concentration. The higher was the ORAC assay result of the extract, the lower was the hexanal concentration in the emulsion after storage.

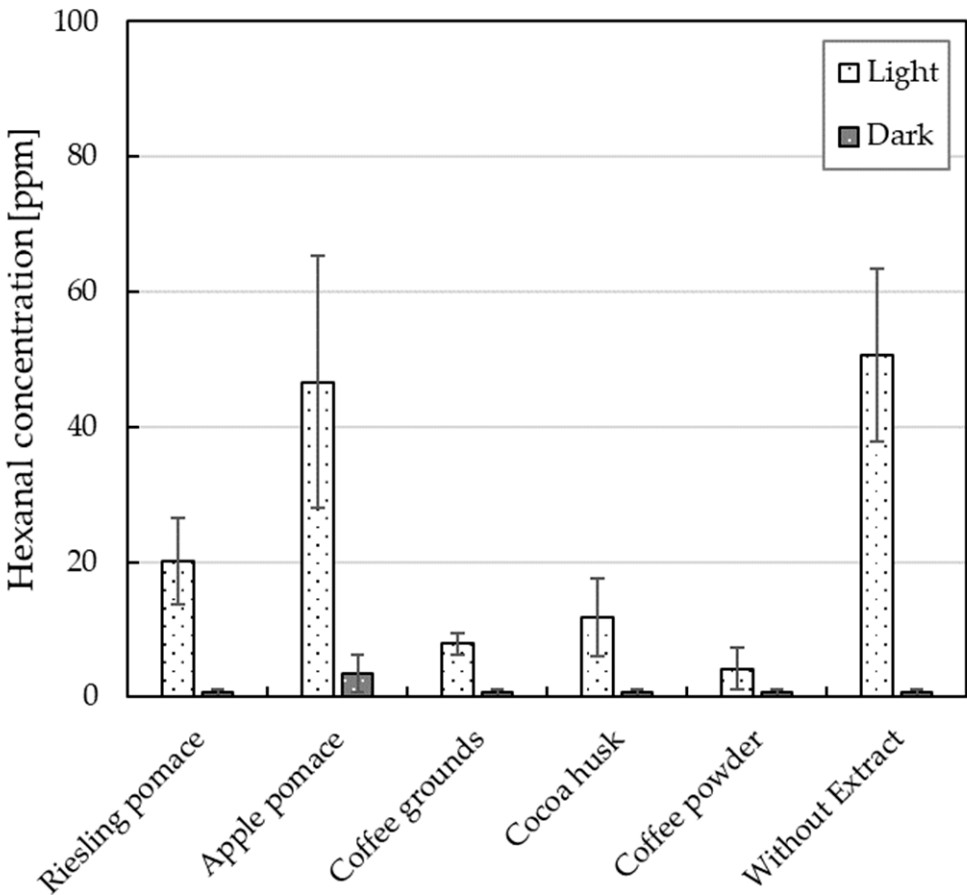

**Figure 7.** Concentration of hexanal in O/W emulsions with plant extracts. The figure shows the concentration of hexanal in mg/L as a fat oxidation product after a storage time of 90 days for the O/W emulsions with and without extract in light and darkness, respectively, as mean values (*n* = 3) with standard deviation (SD).

**Table 6.** *p*-Values of all pairwise multiple comparison procedures (Holm–Šidák method) of the concentration of hexanal in O/W emulsions with plant extracts (significance *** $p < 0.01$).

| | | Riesling Pomace | | Apple Pomace | | Coffee Grounds | | Cocoa Husk | | Coffee Powder | | Without Extr. |
|---|---|---|---|---|---|---|---|---|---|---|---|---|
| | | Light | Dark | Light | Dark | Light | Dark | Light | Dark | Light | Dark | Light |
| Riesling pomace | Light | | | | | | | | | | | |
| | Dark | 0.118 | | | | | | | | | | |
| Apple pomace | Light | 0.006 *** | <0.001 *** | | | | | | | | | |
| | Dark | 0.304 | 1.000 | <0.001 *** | | | | | | | | |
| Coffee grounds | Light | 0.847 | 1.000 | <0.001 *** | 0.997 | | | | | | | |
| | Dark | 0.111 | 1.000 | <0.001 *** | 1.000 | 0.999 | | | | | | |
| Cocoa husk | Light | 0.998 | 0.917 | <0.001 *** | 0.997 | 1.000 | 0.898 | | | | | |
| | Dark | 0.109 | 1.000 | <0.001 *** | 1.000 | 0.999 | 1.000 | 0.911 | | | | |
| Coffee powder | Light | 0.379 | 1.000 | <0.001 *** | 1.000 | 1.000 | 1.000 | 0.999 | 1.000 | | | |
| | Dark | 0.116 | 1.000 | <0.001 *** | 1.000 | 0.999 | 1.000 | 0.922 | 1.000 | 1.000 | | |
| Without Extract | Light | 0.001 *** | <0.001 *** | 1.000 | <0.001 *** | <0.001 *** | <0.001 *** | <0.001 *** | <0.001 *** | <0.001 *** | <0.001 *** | |
| | Dark | 0.113 | 1.000 | <0.001 *** | 1.000 | 0.999 | 1.000 | 0.905 | 1.000 | 1.000 | 1.000 | <0.001 *** |

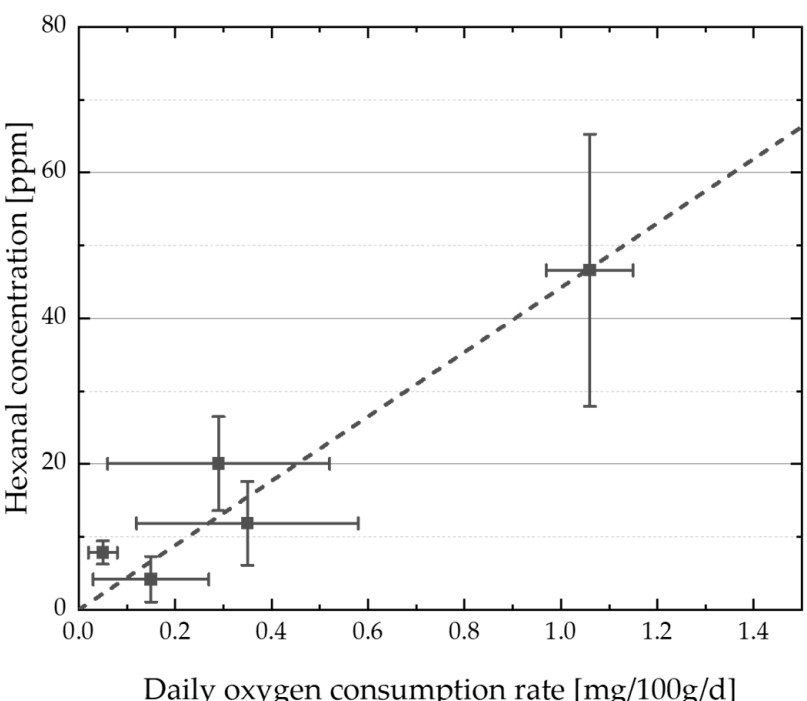

**Figure 8.** Correlation between hexanal concentration and daily oxygen consumption rate of O/W emulsions with plant extracts. The figure shows the hexanal concentration in mg/L as a function of the daily oxygen consumption rate in mg $O_2$ per 100 g per day for the emulsions with extract, as mean values (*n* = 3) with linear equalization function ($R^2$ = 0.99) for all samples stored in light.

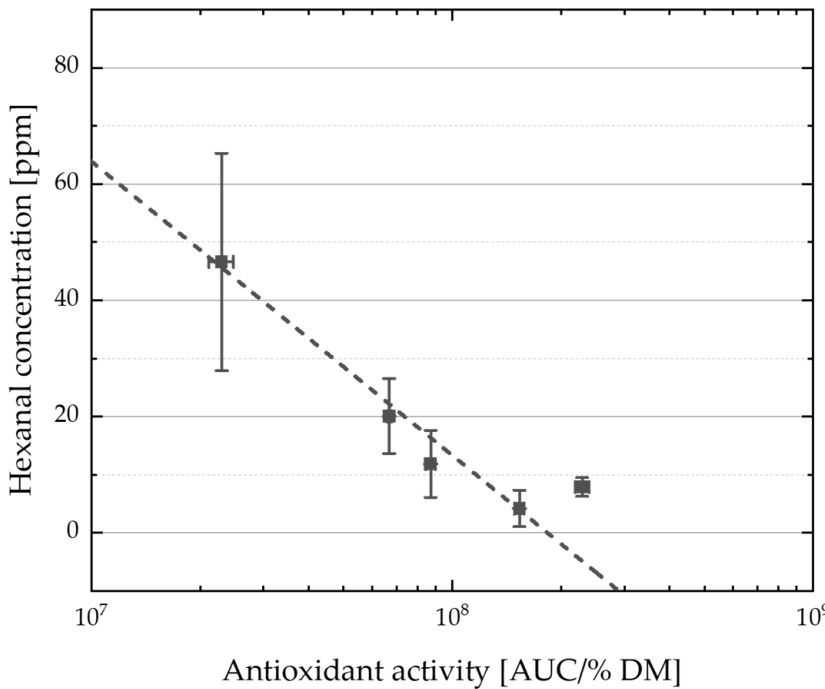

**Figure 9.** Correlation between hexanal concentration and antioxidant activity of O/W emulsions with plant extracts. The figure shows the hexanal concentration in mg/L for the emulsions with extract as a function of the antioxidant activity for all extracts as mean values (*n* = 3) with exponential equalization function ($R^2$ = 0.96) for all samples stored in light.

## 4. Discussion

### 4.1. Antioxidant Activity Results

The highest antioxidant activity in the plant extracts (Section 3.1) was measured with both coffee samples. The antioxidant effect of coffee is associated with several contained hydroxycinnamic acids [54,55]. In addition, caffeine and trigonelline exhibit antioxidant activity [56]. Roasting additionally produces phenyl alanines, melanoidins (brown pigments), heterocyclic compounds, and Maillard reaction products, which are also highly potent antioxidants [55,57–60]. Brewing coffee powder should actually decrease the content of water-soluble antioxidants in the coffee grounds. The reason for the higher value of our coffee grounds probably lies in the type of coffee used, which naturally had a higher content of polyphenols. Cocoa flavanols consist mainly of monomeric (+)-catechin and (−)-epicatechin and oligomeric flavanols (procyanidins) ranging from dimers to decamers [61–63]. Catechins and procyanidins are also present in white grapes [64] (together with several tartaric acids) and apple pomace (in addition to quercetin derivatives, dihydrochalcones, and several hydroxycinnamic acids) [65–67]. All of these substances are polar and are known to have antioxidant activity due to their chemical structures [46,62,68–71]. The total antioxidant content was slightly higher in coffee than in cocoa, and coffee also showed a significantly higher in vitro effect compared with white wine, which is consistent with our results [72].

When using plant extracts in preserved emulsions, it must be noted that a change in pH is possible depending on the composition of the extract. It is therefore important for good comparability to choose a preservative with a broad pH range (e.g., ethanol [47,48]). A subsequent adjustment of the pH value would falsify the comparability between the samples due to the technological steps required for this, such as homogenization and the necessary additional ingredients (acid, base, buffer).

However, challenges should be taken into account especially with by-products of the food industry as a resource for cosmetic extracts. For example, to implement apple pomace, disadvantages may occur due to the different polyphenol contents throughout the apple geographical region of growing, harvest periods, climate changes, varieties, and the different preservation and extraction methods, the standardization of the whole process, which should be faced. Additionally, apple pomace has a low microbial stability; that is why a mild preservation of the by-product immediately after leaving the food supply chain also needs to be implemented in order to guarantee a safe raw material. In addition to that, a regular supply cannot be guaranteed as production and farming volumes of apples and their antioxidant potential can vary from year to year and between different producers. Nevertheless, the same problems occur at the valorization of any fruit or vegetable by-product, such as olive mill waste. However, the opportunities of a green production and circular economy and the functional properties and the increasing demand on natural products dominate the decision making, whether food by-products should be reused or not. Even though apple pomace is still not widely used for cosmetic products, other by-products of the food industry found their way into the cosmetic industry by conquering similar hurdles [73,74]. As the concentration of substances in natural extracts may vary, it is important to test the influence of the extracts on oxidation in practice. Based on these results, future work should focus on the occurrence of pro-oxidative effects by optimizing the application concentration of the extracts and finding a formation of synergies through the combination of plant extracts in order to be able to achieve optimal shelf life and stability in emulsions.

### 4.2. Oxidation Kinetics

The oxidation parameters oxygen uptake (Section 3.2) and daily oxygen consumption rate (Section 3.3) showed particularly strong increases in the samples with apple pomace and without extract. The particularly high increase in emulsion samples with apple pomace during storage in the light can be explained by the presence of photosensitizers, such as chlorophyll, carotenoids, and pro-oxidative polyphenols [75,76]. This transfers the light

energy to an activated oxygen molecule, which can then oxidize the fatty acids more quickly [77,78]. As soon as the oxygen in the headspace has been completely consumed, a plateau is approached, and the uptake does not increase any further. It was observed, especially with the apple pomace sample stored in light, that the standard deviation decreased after reaching the plateau. We suggest that, at this point, a uniform equilibrium has formed in the triple determination, and the variations were probably caused by the sensors.

Daily oxygen consumption showed a higher standard deviation for the samples with Riesling pomace, cocoa husk, and especially the sample without extract stored under light exposure. The high standard deviation during the oxidation process might be explained by the influence of inhomogenities in the matrix and differences in the reaction rate. The samples have already been filled before storage. Despite the greatest amount of accuracy, we assume that it was not possible to fill all samples identically into the vials and to treat them identically during storage. It is conceivable that microscopic air inclusions, temperature differences, and light reflections might lead to local minimally inhomogeneous conditions that affect oxidation.

The course of oxygen uptake was observed in this case to be linear, although an exponential course is often described in the literature, especially after the initial phase of the radical chain reaction [11]. We assumed that, in this case, the limiting factor might be the subsequent transport of oxygen by diffusion through the water phase. The ingredients in the water phase can have an influence on the polarity and oxygen solubility, for example, ethanol [79,80]. Furthermore, we propose to systematically examine the diffusion rate of oxygen in emulsions depending on the ingredients in the water phase in the following work.

In general, a positive slope of oxygen uptake indicates the occurrence of oxidative processes. A negative slope and therefore a decreasing oxygen uptake in the oxidation kinetic graphs may be a sign of oxygen release from the emulsion into the headspace of the vial due to the solubility of the oxygen or microscopically small air bubbles, which could not be prevented by evacuating the bulk. Similar challenges measuring the oxidation of emulsions in vials have already been described in the literature [48].

As described in a previous paper, the consideration of oxygen uptake and the daily oxygen consumption rate measured by optical sensors for the oxidation of emulsions offers both benefits and limitations. The course of the oxygen uptake well depicts the oxidation process, but might not always be comparable between samples or measurement series, especially when implementing parameters are changed. Meanwhile, the daily oxygen consumption rate provides a single comparable value, but its calculations include points before the obvious initiation of oxidation, which affects the standard deviation and statistical analysis [48]. Again, the values are only comparable within experimental cases that were performed identically. The optical sensors also offer disadvantages, for example, due to the degradation of the indicator in light or an interaction with ingredients in the sample, such as acetone. However, the sensors placed in the samples stored under light exposure were covered to prevent light degradation, and no cross-sensitivity was specified by the manufacturer for the used ingredients, including ethanol [81]. At the end of the storage, it was observed that the oxygen sensors were intact and provided realistic results. Nevertheless, it might be beneficial to consider other measurement methods for examining oxygen kinetics of oxidation processes in emulsions in the future. Measurement of the headspace above the samples would also be possible using sample gas analyzers. The oxygen can be measured electrochemically via a zirconium oxide electrode. The incoming gas mixture from the sample is compared with the reference air (room air), so the oxygen concentration difference can be determined by the conductivity of the zirconium grid. Zirconium oxide is a permeable membrane only for oxygen, which makes the measurement highly selective [82]. This measurement has already been successfully described for emulsions in the literature [83]. However, these measurements are invasive; thus measurement of one sample over multiple measurement dates would not be possible. In contrast, pressure sensors can measure the concentration of oxygen in a noninvasive and automated way.

It has been described that the binding of oxygen by lipid peroxidation leads to a change in pressure and can be measured accordingly with pressure sensors [84]. We propose to investigate the suitability of this analysis for measuring oxidative emulsion stability.

Regardless of the quantification method, the oxygen uptake provides the amount of oxygen bound, but there is no information on whether this oxygen has led to a degradation in quality. Therefore, the addition of a measurement of oxidation products, such as hexanal, to the study proved to be suitable. Oxygen uptake can reflect the oxidation of a product well if other factors, such as microbial growth, are excluded [11,17,85]. However, microbial metabolites, such as $CO_2$, were below the detection limit of 0.1% in the headspace gas analysis in all vials after storage, which indicates that the possibly existing microorganisms have not multiplied. Therefore, it can be concluded that the samples were sufficiently preserved and oxygen was not consumed by respiring microorganisms [86].

*4.3. Hexanal Concentration*

High hexanal concentrations could indicate very strong oxidation and orthonasal perceptible off-flavor. In previous studies, a rancid off-flavor was measured above the concentration of 0.32 mg/L in oil [11]. Other studies described a rancid odor threshold for hexanal concentrations above 5 to 10 mg/L in low-fat foods [87]. In our study, the samples with Riesling pomace, apple pomace, and cocoa husk extract showed hexanal concentrations above those limits, indicating a rancid scent, which could be a risk for the acceptance of the product (Section 3.4). The daily oxygen consumption rate and the hexanal concentration seemed to correlate. The headspace measurement of the samples and the calibration did not seem to be influenced by matrix effects. While at the beginning of the chromatogram the ethanol peak might lead to a more difficult evaluation of substances in close proximity due to its height and width, this did not collide with the evaluation of hexanal, which was clearly less volatile and eluted later. To be on the safe side, a prior separation of the ethanol might have to be considered, which in turn involves the risk of losing relevant substances.

Calibration was performed for values between 1 and 6 mg/L, which are within the range of the human perception threshold and in the range of the acceptance limit. It is conceivable that the calculated value was not completely accurate for heavily oxidized samples, since their concentration was beyond the calibration range. However, these concentrations are far above the acceptable range, and it is assumed that these samples would be rejected by the consumer regardless of the exact value. The calibration was performed in matrix on one sample. Since the matrix is identical in all samples except for the composition of the plant extract dry matter, the calibration should be transferable between samples. For certainty, the standard addition to each sample and a separate quantification in each matrix individually is recommended. In general, hexanal content is not an indicator of the level of oxidized fatty acids in relation to all fatty acids contained, but in relation to the total mass of the product. This could be a relevant aspect if the oil content of the emulsion changed, for example, when analyzing W/O emulsions. It should be taken into account that hexanal may not be formed in all fatty acids and not in all oxidation reactions [11].

Compared with the quantifications for oxygen, the hexanal concentration measurement provided visibly lower standard deviations. The authors suggest several reasons, therefore. On the one hand, hexanal was assumed to be predominantly present dissolved in the matrix, and its concentration is less affected by weather-dependent fluctuations in atmospheric pressure since the conditioning parameters for GC analysis were standardized. Additionally, the measurement methods and matrix-inclusive calibration were very precise. On the other hand, the measurement was performed at the end of the storage period, which means that the reaction time was sufficient to produce the measured amount of hexanal. In numerous studies, aldehydes as hexanal are regarded exclusively as an oxidation parameter, which is transferable to this approach [88,89]. However, it should be considered that hexanal can be further oxidized to hexanoic acid [90]. Therefore, measurements of hexanoic

acid may be taken into account for future work to confirm the suitability of hexanal as an oxidation parameter in the particular approach.

*4.4. Correlations*

The daily oxygen consumption rate was highest in the samples that showed the highest increase in oxygen uptake and generally higher in the samples stored under light exposure, indicating a higher oxidation when exposed to light. The hexanal concentration also behaved in a similar way. The highest values for both parameters, daily oxygen consumption rate and hexanal concentration, were measured in samples with apple pomace. In both parameters, the emulsions with Riesling pomace, coffee grounds, cocoa husk, and coffee powder were also comparable. Only the sample without extract showed a similar high level of hexanal to apple pomace, while having a daily oxygen consumption rate similar to the other variants. One possible explanation is the oxidation of ingredients from the apple extract, which absorb oxygen to a high degree but do not lead to fat oxidation and the formation of hexanal to the same extent as the blank sample. Another option would be an altered fat oxidation process due to photosensitizers from the apple extract that affected the reaction mechanisms and led to other reaction products where hexanal was not formed.

However, samples without extract could not be compared with those with extract for the correlation analysis. We assumed that, in the emulsions with extract, the antioxidants were targeted first during oxidation and stabilized the radicals. When these were consumed, then the fatty acids reacted to form peroxides and later volatile substances, such as hexanal. In emulsions without extract, the unsaturated fatty acids were targeted first. Therefore, both were not comparable.

In general, the daily oxygen consumption rate in light correlated with the ORAC assay results. One explanation factor could be specific oxidation processes in light, which might be represented by the reactions in the assay. The ORAC assay maps the radical scavenging properties as a color reaction, which is evaluated photometrically as a peak integration area under the curve. The more radicals a molecule can retain, the higher the assay result [52,53]. Since the radical chain reaction, which is represented by the daily oxygen consumption rate, is described as exponential when uninhibited [11], an exponential correlation was found especially in those samples with low antioxidant activity. The linear relationship between daily oxygen consumption rate and hexanal concentration is likely to be causal, since the uptake of oxygen leads to the formation of fatty acid peroxides and subsequently to decomposition, which can produce hexanal [91–93]. This was described not only in oils, but also in the degradation of lipid peroxides from autoxidation in aqueous systems [94]. The exponential correlation between antioxidant activity and hexanal formation can be explained by the fact that the reaction to hexanal is directly dependent on oxidation, and oxidation is dependent on antioxidant activity. This is therefore a sequence of causality in these processes.

The correlations suggest that for the O/W emulsions used, the oxygen consumption rate during storage may be estimated from the antioxidant activity of the extract used, and the hexanal concentration could be extrapolated from the oxygen consumption rate. It is also possible to directly conclude the hexanal concentration from the antioxidant activity measured by ORAC assay in case of matrices that are produced and stored identically to the samples used in our approach. By combining the correlation between hexanal concentration and oxygen uptake with organoleptic (orthonasal) analysis by a trained panel, the maximum tolerable amount of oxygen can be determined, and from this, the best-before date can be calculated [95]. Therefore, the discovered correlations put in perspective estimating shelf-life models and minimum barriers in the product packaging in the future.

*4.5. The Role of the Emulsion on Oxidation*

Various factors may play a role on the oxidation in O/W emulsion matrices, some of which have already been discussed in the literature, such as pH value and emulsifier type [96,97]. Due to the micellar structure, the oil may have been protected in the O/W

emulsions, since the oil phase was surrounded by the water phase with solved polar antioxidants [98]. On the other hand, the large surface area of the fat micelles could enable a reaction on the interface with oxygen dissolved in water, as was described already for the impact of micelle sizes in food emulsions [99,100]. The water-soluble antioxidants from the extracts could scavenge oxygen already in the aqueous phase, as described already for ascorbic acid [101]. However, the major water phase might have diluted water-soluble antioxidants, which could impact their antioxidant activity, which was also described as concentration dependence for polyphenols in wine [102]. In addition, fat-soluble photosensitizers, such as chlorophyll from apples [103], may have built up in the oil phase and led to an increased oxidation. These and other pro-oxidants could have an impact here. Among the most important antioxidants are phenolic compounds, which can have a pro-oxidant effect as well, such as flavonoids, ascorbic acid, $\alpha$-tocopherol, and carotenoids [104]. Conditions such as an acidic pH, metal ions, certain concentrations of substances, and/or high oxygen partial pressure can trigger pro-oxidant effects between the ingredients [76,105]. Therefore, it is important to choose the ingredients in the right ratios suitable for the emulsion type. It might be useful to systematically study the reaction mechanisms during oxidation in emulsions depending on the ingredients used in the future.

Although use concentrations of plant extracts in the cosmetic industry are often given as the proportion of the liquid extract to the total bulk, we have chosen to use the dry matter in the interest of comparability between different extracts and correlatability with the analysis data. The input concentrations in Table 2 show that the concentration of the liquid extract was comparable to or even higher than typical formulations of emulsions. However, the active ingredient content was assumed to be comparable due to the calculation in relation to the dry mass. Nevertheless, normalization to dry matter might have the disadvantage that volatiles could be missed in the dry matter analysis procedure and were not included in the calculation.

The results of this study are only applicable to the investigated emulsion type (O/W). However, due to the reversal of the phases and the higher proportion of the oil phase, the oxidative processes in W/O emulsions and their quantification by instrumental analysis may behave differently. In contrast to O/W emulsions, where the oxygen transport through the water phase was the limiting factor for oxidation of the dispersed oil phase, the oxidation rate in W/O emulsions might be increased due to direct oxygen contact on the surface of the lipid phase [19]. Additionally, the solubility of oxygen in the continuous oil phase increases with temperature, according to the Bunsen coefficient [79,80]. Depending on the plant-based oil used, the changed oil-to-water ratio results in a higher concentration of unsaturated fatty acids and, therefore, might lead to a higher risk of rancid volatile. In the meantime, due to their lipophilic character, their solubility and dilution in the oil phase might also lead to a decrease in perceptible off-flavors [11]. In the future, we propose to examine the antioxidant influence of ingredients also on W/O emulsions analogous to this approach.

## 5. Conclusions

The aim of this study was to determine the suitability of a method combination by measuring the antioxidant activity, oxygen concentration, and volatile oxidation products via GC for the characterization of the influence of some plant extracts on the oxidative stability of O/W emulsions. This was achieved by measurements of the antioxidant activity of the extracts via ORAC assay, the oxygen consumption rate of the emulsions, and the quantification of hexanal via HS GC–MS during accelerated storage tests with and without light exposure. The results showed that all emulsion samples stored in darkness showed insignificant changes, whereas under light exposure, a preferential oxidation occurred due to a higher oxygen uptake and an increased formation of characteristic fatty acid oxidation products (hexanal). Depending on the incorporated plant extract, significant differences in oxygen uptake behavior were observed under the influence of light that indicate possible antioxidative effects. In this use case, the results showed also a mathematical relationship

between ORAC assay results, daily oxygen consumption rate, and hexanal concentration of O/W emulsions during storage. These results show promising perspectives for the research areas of shelf life modelling or modelling-assisted product development.

**Author Contributions:** Conceptualization, all authors; methodology, H.Z., A.S.; formal analysis, all authors; investigation, H.Z.; data curation, H.Z., A.S.; writing—original draft preparation, A.S.; writing—review and editing, A.S., H.Z., K.B.; supervision: K.B. All authors have read and agreed to the published version of the manuscript.

**Funding:** This research received no external funding.

**Institutional Review Board Statement:** Not applicable.

**Informed Consent Statement:** Not applicable.

**Data Availability Statement:** All data sources are listed in the references.

**Acknowledgments:** The present work was performed in partial fulfillment of the requirements for the Master of Science in Engineering degree at Management Center Innsbruck (MCI) for H.Z., and for the "Dr. rer. biol. hum." degree at FAU Erlangen-Nürnberg for A.S. We would also like to thank MCI for making this master's thesis possible. We thank the Process Development for Plant Raw Materials team for their great support in the preparation, analysis, and evaluation of the extracts as well as statistical and correlation analysis. Furthermore, thanks are due to Christine Berger and Christopher Schmidt for their experimental support of this study in the form of physicochemical analysis. Additionally, we would like to thank our colleague Matthias Reinelt, who provided helpful advice with his experience and expertise in the correlation analysis of oxidative processes. We also appreciate Frank Welle's proofreading and support.

**Conflicts of Interest:** The authors declare no conflict of interest.

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
