# Peer review of "The Influence of Antioxidant Plant Extracts on the Oxidation of O/W Emulsions"

_cosmetics, doi:10.3390/cosmetics10020040_

Round 1
Reviewer 1 Report
In this work, a type of plant-based antioxidant extract was used, then investigated the influence on the oxidative stability of natural cosmetic emulsions. The current manuscript needs minor revision before acceptance and some recommendations are listed.
1. There are some reference errors that occurred, such as lines 246, and 296. Please check the whole manuscript carefully.
2. In Figure 4, due to the error bar, some lines overlap. Does it still get the conclusion that the oxidation was more intense with apple pomace than in the other extracts?
Author Response
Thank you for your effort and time. We appreciate your positive feedback and helpful suggestions. We have checked the sources again and made additions. We have revised our assessment of the oxidation process in the dark. Visually, the oxidation progression apple extract might have appeared higher than the others, but the changes were very small and the subsequent statistical analysis showed no significant change.
Reviewer 2 Report
Dear Authors,
I write you in regard to the manuscript entitled entitled "Measurement of the Influence of Antioxidant Plant Extracts on the Oxidation of O/W emulsions".
- please, add to the abstract the plants.
- add botanical names in all samples.
- justify the use of the ethanol in the emulsion and described the quantitative composition including all ingredients.
- why was the concentration of actives that low?
- text presented several typos.
- what would explain the high values os SD in figure 5?
Author Response
Dear Reviewer, you will find our response attached in the document.
Reply Letter for Reviewer 2
Dear Reviewer,
Thank you very much for your effort and time to evaluate our manuscript. Your comments were very helpful to further improve the paper. In addition, we have adjusted the heading, as we have noticed that in this work the focus is not only on the measurement but on the correlating relationship of the data and in general on the influence of the parameters on each other. In the following we would like to respond to your comments.
- please, add to the abstract the plants.
- add botanical names in all samples.
We adjusted the abstract and the text passages with the plant names.
- justify the use of the ethanol in the emulsion and described the quantitative composition including all ingredients.
We have explained the formulations in Table 1 and 2 and explained the use of ethanol in the text. In previous studies, ethanol showed the least oxidative interaction with the formulation and good preservative qualities, which is why it is in our opinion suitable to study the oxidative influences of the extracts.
- why was the concentration of actives that low?
Although use concentrations of plant extracts in the cosmetic industry are often given as the proportion of the liquid extract to the total bulk, we have chosen to use the dry matter in the interests of comparability between different extracts and correlatability with the analysis data. The input concentrations in Table 2 show that the concentration of the liquid extract is comparable to typical formulations of emulsions. However, the active ingredient content were the same due to the calculation in relation to the dry mass. The ethanol and water content was adjusted based on the extract input. We discovered that there was a mix-up of units in the concentration information and are very grateful that we were able to correct this (it was 0,27 g dry mass per 150 g sample, not 0,27 %). In addition, we have defined a specification range in which the final concentration of the extract solids should lie (0.16-0.18 %). We revised the manuscript accordingly.
- text presented several typos.
We revised the manuscript and hope to have found all errors.
- what would explain the high values of SD in figure 5?
Daily oxygen consumption showed a higher standard deviation for the samples with Riesling pomace, cocoa husk and especially the sample without extract stored under light exposure. The high standard deviation during the oxidation process might be explained by the influence of inhomogenities in the matrix and differences in the reaction rate. The samples have already been filled before storage. Despite the greatest amount of accuracy, we assume that it was not possible to fill all samples identically into the vials and to treat them identically during storage. It is conceivable that microscopic air inclusions, temperature differences and light reflections might lead to local minimally inhomogeneous conditions that affect oxidation. We revised the manuscript accordingly in the discussion.
Round 2
Reviewer 2 Report
Dear Authors,
Sample composition did not match the text and table.
Author Response
Dear Reviewer, thank you very much for the advice. The % mentioned was given as a total value for the respective phase. Of course, it is better to mention the individual ingredients in the text. We have adjusted the manuscript accordingly.